# Incidence and transmission of respiratory syncytial virus in urban and rural South Africa, 2017-2018

Cheryl Cohen [1,2] ✉, Jackie Kleynhans [1,2], Jocelyn Moyes[1,2], Meredith L. McMorrow[3,4], Florette K. Treurnicht[1,5], Orienka Hellferscee [1,5], Nicole Wolter [1,5], Neil A. Martinson[6,7,8], Kathleen Kahn [9], Limakatso Lebina[6], Katlego Mothlaoleng[6], Floidy Wafawanaka[9], Francesc Xavier Gómez-Olivé [9], Thulisa Mkhencele[1], Azwifarwi Mathunjwa[1], Maimuna Carrim [1,5], Angela Mathee[10], Stuart Piketh [11], Brigitte Language[11], Anne von Gottberg [1,5] & Stefano Tempia [2,4,12]

Data on respiratory syncytial virus (RSV) incidence and household transmission are limited. To describe RSV incidence and transmission, we conducted a prospective cohort study in rural and urban communities in South Africa over two seasons during 2017-2018. Nasopharyngeal swabs were collected twice-weekly for 10 months annually and tested for RSV using PCR. We tested 81,430 samples from 1,116 participants in 225 households (follow-up 90%). 32% (359/1116) of individuals had ≥1 RSV infection; 10% (37/359) had repeat infection during the same season, 33% (132/396) of infections were symptomatic, and 2% (9/396) sought medical care. Incidence was 47.2 infections/100 person-years and highest in children <5 years (78.3). Symptoms were commonest in individuals aged <12 and ≥65 years. Individuals 1-12 years accounted for 55% (134/242) of index cases. Household cumulative infection risk was 11%. On multivariable analysis, index cases with ≥2 symptoms and shedding duration >10 days were more likely to transmit; household contacts aged 1-4 years vs. ≥65 years were more likely to acquire infection. Within two South African communities, RSV attack rate was high, and most infections asymptomatic. Young children were more likely to introduce RSV into the home, and to be infected. Future studies should examine whether vaccines targeting children aged <12 years could reduce community transmission.

In 2019, respiratory syncytial virus (RSV) was estimated to cause 33 million acute lower respiratory tract infections (ALRI) globally. Among children <5 years of age, there were an estimated 3·6 million ALRI hospital admissions and >100,000 deaths; approximately 2% of all-cause deaths in children aged 0–60 months. The majority of the burden ( > 97%) occurred in low- and middle-income countries (LMIC), and approximately three quarters were medically unattended[1]. The incidence of RSV-associated ALRI is greatest among infants aged <6 months; however, an estimated 80% of RSV-associated ALRI among children aged <5 years occurs in children aged 6 months to 4 years. There is also a substantial burden of severe RSV among older adults[2].

Interventions to prevent severe RSV among infants (long-acting monoclonal antibodies and maternal vaccination) are at an advanced stage of development, with one product licensed by the European

Medicines Agency in 2022 and others anticipated in 2023[3,4]. Live-attenuated vaccines targeting children aged >6 months are also under development. In addition to protecting against illness beyond the first 6 months of life, vaccination of children aged >6 months has the potential to reduce RSV transmission, leading to overall reductions in disease burden and reduced transmission to vulnerable infants and older adults through indirect effects. To project the potential benefits of RSV prevention interventions on virus transmission, it is essential to understand the risk of RSV infection and patterns of transmission in communities.

Data on RSV incidence and household transmission from sub-Saharan Africa are limited. An intensive study of RSV transmission in Kenya conducted in the 2009-2010 RSV season found high RSV attack rates (>60%) among infants and that older children in the household were responsible for approximately three-quarters of introductions resulting in transmission to infants[5]. This study was limited to a single RSV season in a rural Kenyan community and included <50 households with an infant. Data from additional areas in sub-Saharan Africa, spanning multiple RSV seasons and including a representative sample of households, are needed.

In a prospectively followed, randomly selected household cohort, we measured the community burden and transmission of RSV in a rural and an urban setting in South Africa from 2017-2018, and reported factors associated with infection and transmission, the symptomatic fraction, and the role of asymptomatic illness in transmission.

## Results

### Study population
We approached 670 households, of which 287 (43%) had >2 household members, and the head of household agreed to participate (Supplementary Fig. 3). Of these, 80% or more of individuals consented in 225 households (78%) and were included in the study. Of 1176 household members, 1116 (95%) were included in the analysis. Each year, different cohorts of individuals were enroled with 558 individuals from 108 households followed up in 2017 and 558 individuals from 117 households in 2018. Reasons for exclusion from analysis for the remaining 60 individuals included relocation, death or ≤10 swabs collected because of refusal or withdrawal of consent. There was a median of 5 household members and 2 sleeping rooms per household, and 68% included a child aged <5 years, with a higher percent in the rural site (p < 0.001) (Supplementary table 2). Participants from the rural site were younger, had a lower education level, and were less likely to be employed. Tuberculosis and underlying illness were less commonly reported in the rural site, and HIV prevalence was higher in the urban site (19%, 99/522 vs 12%, 68/553).

### Follow up
Out of 90,041 potential twice-weekly follow-up visits from January through October each year, we collected and tested 81,430 (90%) nasopharyngeal swabs, of which 796 (1%) tested positive for RSV on rRT-PCR (Figs. 1, 2 and Supplementary Fig. 4). Reasons for not collecting a swab during the 8611 visits were participant traveling (n = 4177), missed visit (n = 4070), or not specified (n = 364).

### Incidence of infection and illness
Overall, 75% (168/225) of households had at least one individual testing RSV-positive each year (Supplementary Table 3). The incidence estimates of RSV infection and illness (at least one symptom) were 47.2 and 15.7 per 100 person-years, respectively but varied by site and year (Fig. 3, Supplementary Table 4). Incidence of infection (irrespective of symptoms) was highest among children aged <1 and 1-4 years (72.6 and 79.1 per 100 person-years, respectively) and lowest among individuals aged ≥65 years (17.8 per 100 person-years, Fig. 4 panel a, supplementary table 4 and supplementary Table 5). Incidence of illness (≥1 symptom) was highest in individuals aged <1, 1-4, and 5-12 years and

adults aged ≥65 years (48.4, 38.3, 14.9, and 11.9 per 100 person-years respectively).

### Repeat infections and mixed infections
Among 359 individuals experiencing at least one RSV infection episode, 31 (9%) had a second RSV infection and 3 (1%) had three RSV infections within the same year (Supplementary Table 6). Repeat infections were most common in individuals aged <18 years and ≥65 years (Supplementary Table 4). There were 396 infection episodes in total, four of which had a mixture of RSV subtypes. We identified one co-infection episode with influenza and RSV.

### Index case characteristics
Among 243 individuals who were the index case at least once, 4% (n = 10) were aged <1 year, 27% (n = 65) 1–4 years, 28% (n = 69) 5–12 years, 15% (n = 37) 13–18 years, 14% (n = 35) 19–44 years, 10% (n = 24) 45–64 years, and 1% (n = 2) ≥65 years. The odds of being an index case were higher among individuals aged ≤18 years compared to 19–44 years (Supplementary Table 7). Among 12 RSV infection episodes in infants aged <1 year, the infant was the index case for 10 (Supplementary Table 8).

### Differences by RSV subgroup
Annual rates of RSV infection varied by subgroup. The overall rates per 100 person-years were higher for RSV B (27.8, 95% CI 24.4–31.6) than A (17.4, 95% CI 14.8–20.5) but varied by year (Supplementary Table 9). RSV A was commonest in infants and infection incidence decreased with increasing age, while peak incidence of RSV B was in children aged 1-4 years (Supplementary Table 10 and Supplementary Fig. 5).

### Symptomatic fraction and characteristics of symptomatic individuals
Overall, 33% of infections were associated with ≥1 symptom and 5% with ILI, with a higher proportion of symptomatic infections in individuals aged <5 (50%) and ≥65 (67%) years (Supplementary Table 4, Fig. 4b). The most common symptoms reported among 132 symptomatic episodes were cough (105, 80%), runny nose (91, 69%) and fever (20, 15%). The rate of medically attended RSV-associated illness was 1.1 per 100 person-years and was highest in infants (6.1 per 100 person-years). Among symptomatic individuals, 7% (9/132) sought medical care and 13% (7/55) of those attending school or work reported absenteeism (Supplementary Table 11). On multivariable analysis, factors associated with symptomatic (vs. asymptomatic infection) were age group <1, 1-4, 5-12 and ≥65 years vs. 13-18 years, shedding duration >10 vs. <4 days, and rRT-PCR Ct value < 30 (Table 1).

### Shedding
The mean duration of shedding was 6.5 days (standard deviation 6.5, range <1-50 days); 3% (12/400) of episodes shed for >21 days (3 aged <1, 4 aged 1-4 and 2 aged 5-12, 2 aged 13-44 and 1 aged ≥65 years). On multivariable analysis, factors associated with longer episode duration were presence of ≥2 vs no symptoms, and rRT-PCR Ct value < 30 (Table 2, Supplementary Fig. 6).

### Household cumulative infection risk
The overall HCIR was 11% (96 of 856 exposed household members) and 61% (147) of 242 clusters included only one individual. Transmission was highest from index cases with ≥2 symptoms (21%, 40 of 193 exposed household members) vs. asymptomatic individuals (8%, 47 of 561 exposed household members) (Table 3). On multivariable analysis controlling for index case age, factors associated with increased transmission were ≥2 symptoms vs. no symptoms and duration of shedding >10 days vs <4 days. Individuals aged 1-4 years vs. ≥65 years were more likely to acquire RSV infection. Among 339 infection episodes in RSV

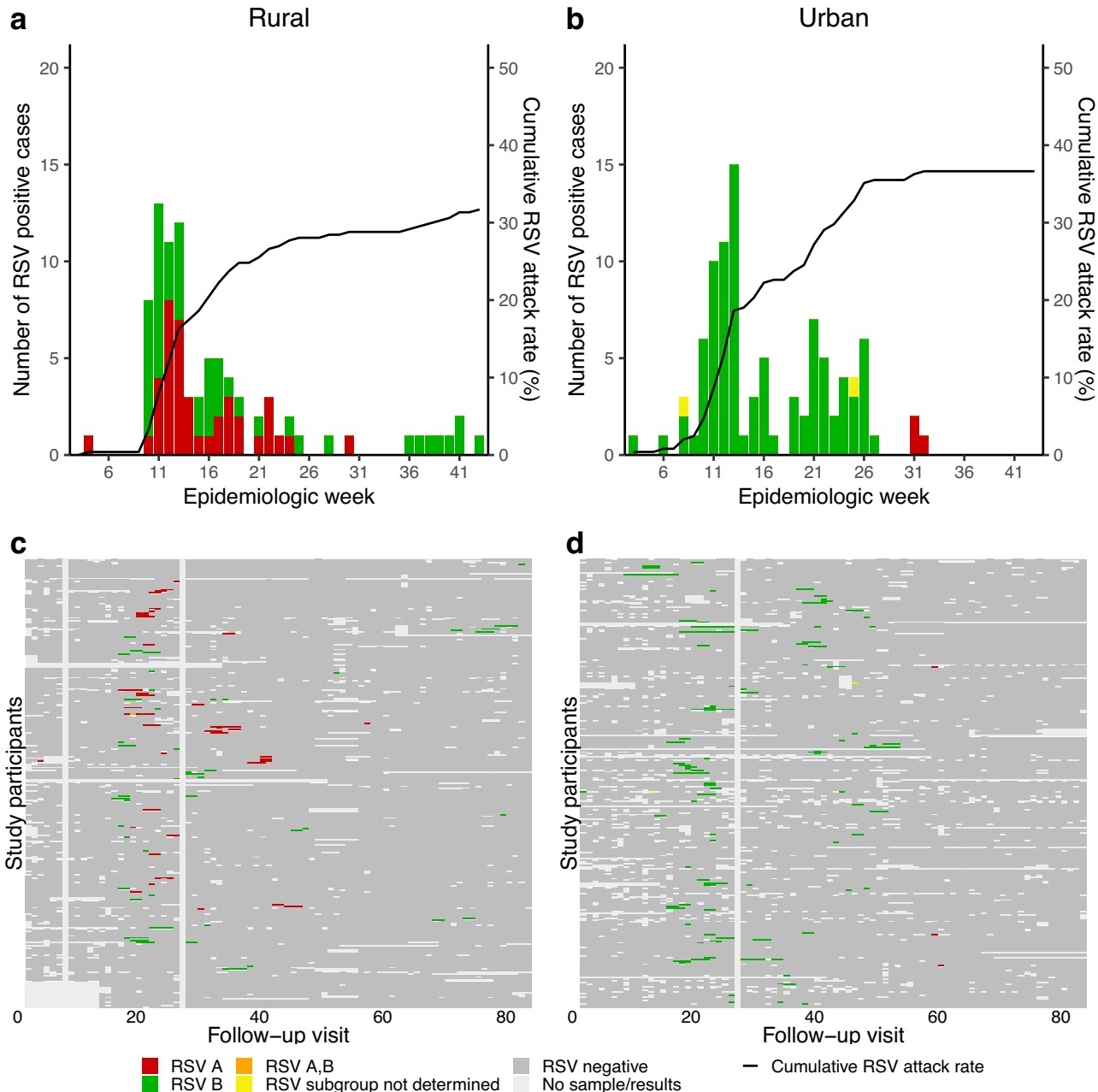

**Fig. 1 | Epidemic curve and results of real-time reverse transcription polymerase chain reaction (rRT-PCR) by study visit at a rural and an urban site, 2017.** Top panel: Number of new cases testing rRT-PCR-positive per study visit and cumulative percentage of individuals infected, **a** a rural site and **b** an urban site, South Africa, 2017. Bottom panel: Results of rRT-PCR of individuals enroled in the PHIRST study, at **c** a rural site and **d** an urban site, South Africa, 2017. Columns are individual follow up visits and rows are individual participants. Individuals within clusters the same household are numbered consecutively (appear below one another). Follow up visits are coloured white if no sample was tested, light grey if the sample tested negative for RSV and coloured red if the nasopharyngeal swab tested positive for RSV A, green if the sample tested positive for RSV B, orange if the sample tested positive for RSV subgroup A and B and yellow if the sample subgroup could not be determined.

clusters without coprimary index cases, 223 (66%) infections were presumed acquired in the community (i.e., were the index case).

### Generation interval
The mean generation interval was 8.4 days (standard deviation 4.0, range 1–16 days) (Supplementary Fig. 7). On multivariable analysis, factors associated with shorter generation interval were index age group <1, 1–4, 5–12, 13–18 and 19–44 years vs. 45–64 years and contact age group 1–4 years vs 5–12 years (Supplementary Table 12). The generation interval was longer for clusters with RSV subgroup B vs. subgroup A.

### Differences between PLWH and HIV-uninfected individuals
On multivariable analysis, after accounting for other factors, when comparing people living with HIV (PLWH) to HIV-uninfected individuals, there were no differences in symptomatic fraction, shedding duration, or probability of transmission or acquisition of infection (Tables 1–3).

### Discussion
In rural and urban South African households, we found that the incidence of RSV infection was high (>45 per 100 person-years) and 10% of infected individuals experienced a repeat infection in the same year.

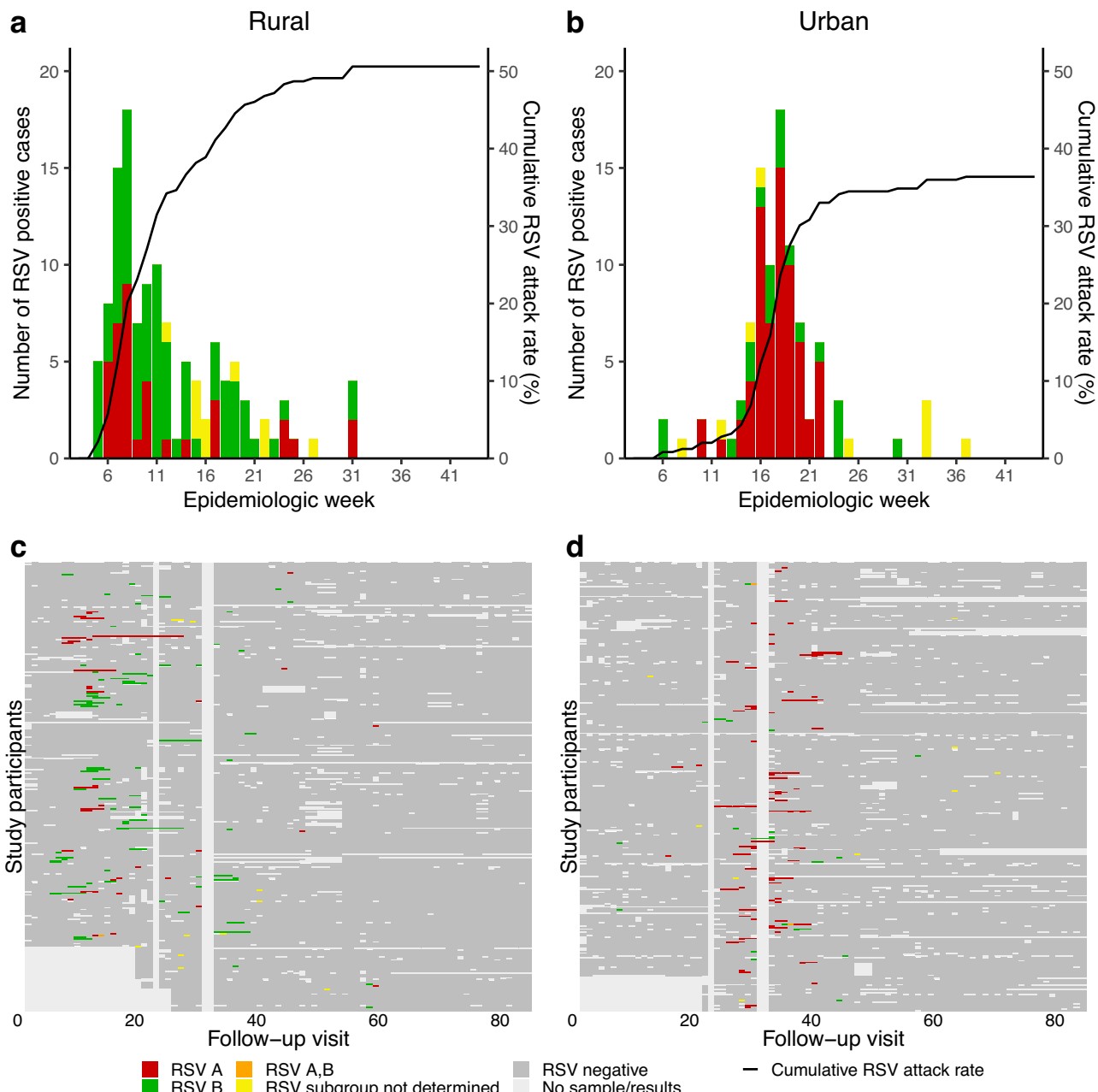

**Fig. 2 | Epidemic curve and results of real-time reverse transcription polymerase chain reaction (rRT-PCR) by study visit at a rural and an urban site, 2018.** Top panel: Number of new cases testing rRT-PCR-positive per study visit and cumulative percentage of individuals infected, **a** a rural site and **b** an urban site, South Africa, 2018. Bottom panel: Results of rRT-PCR of individuals enroled in the PHIRST study, **c** a rural site and **d** an urban site, South Africa, 2018. Columns are individual follow up visits and rows are individual participants. Individuals within the same household are numbered consecutively (appear below one another). Follow up visits are coloured white if no sample was tested, light grey if the sample tested negative for RSV and coloured red if the nasopharyngeal swab tested positive for RSV A, green if the sample tested positive for RSV B, orange if the sample tested positive for RSV subgroup A and B and yellow if the sample subgroup could not be determined.

One-third of infected individuals experienced symptoms, with symptoms more common at the extremes of age. Incidence was highest among children aged <5 years (exceeding 70 infections per 100 person-years); children ≤12 years old accounted for 60% of index cases. These findings, together with the fact that children aged 1-4 years are more likely to acquire infection within the household, suggests that children aged <5 years are important drivers of RSV transmission in the household.

The high attack rates of RSV are similar to previous prospective cohort studies. A cohort study in Kenya conducted with a similar design over a single RSV season, identified at least one RSV infection in 85% of households, an overall RSV infection attack rate of 37%, and 13% of individuals experiencing a repeat infection[6]. A much earlier detailed US study[7], which ascertained the presence of RSV using culture, identified RSV infection in 44% of households and 22% of individuals. The lower attack rates were likely a result of the less sensitive diagnostic approach used. We found that the highest attack rates were in young children, similar to Munywoki et al. where 64% of infants experienced at least one RSV infection[5]. We observed differences in the age-specific attack rates by RSV subgroup with RSV A commonest in infants and RSV B commonest in slightly older children. A previous study from South Africa did not find differences in clinical severity

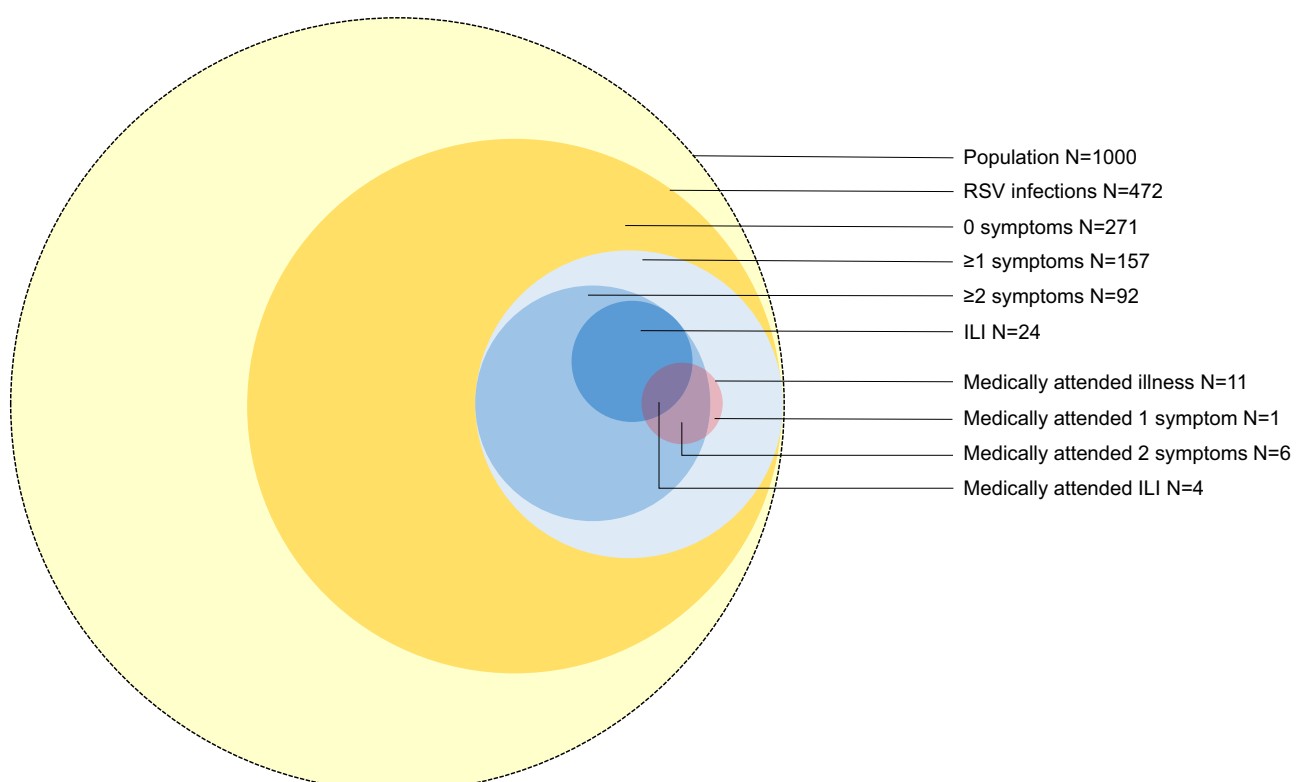

Population N=1000
RSV infections N=472
0 symptoms N=271
≥1 symptoms N=157
≥2 symptoms N=92
ILI N=24
Medically attended illness N=11
Medically attended 1 symptom N=1
Medically attended 2 symptoms N=6
Medically attended ILI N=4

**Fig. 3 | Venn diagram of estimated numbers of RSV infection episodes by symptoms and medical attendance per season in a population of 1000 individuals, a rural and an urban site, South Africa, 2017-2018.** ILI influenza like illness (fever and cough).

between RSV A and B, but some studies have suggested that severity may vary by subgroup[8–11]. We did not find any association with HIV infection status and incidence of infection, symptomatic fraction, or acquisition of infection. An important limitation of this analysis was the relatively small numbers of PLWH included in the study which may have limited power to detect associations. We were also not able to compare the characteristics of PLWH well-controlled on treatment to those poorly controlled.

Despite the high infection attack rates observed, only one in three episodes in our study were associated with symptoms. This is lower than the 58% symptomatic infections in a Kenyan cohort with a similar design[6]. The substantially younger study population in the Kenyan study could have contributed to a higher symptomatic fraction as symptoms were commonest in younger children. It is possible that some individuals in our study may not have reported very mild symptoms, particularly as we followed individuals for 10 months, potentially leading to reporting fatigue. We attempted to minimize non-reporting by systematically asking participants about the presence or absence of symptoms at each visit, conducting monthly field worker training on symptom data collection, and reiterating to household members the importance of reporting all symptoms at each visit. We found that individuals at the extremes of age were more likely to report symptoms, similar to Munywoki et al. and in keeping with the fact that these age groups are at highest risk of severe illness[1,12,13]. The commonest reported symptom was a cough, reported in 80% of symptomatic individuals, with fever relatively uncommon, reported in only 15% of cases. This confirms that case definitions used in surveillance, such as ILI which require fever and cough, will miss a large proportion of cases[14]. A limitation of our study was that we did not collect data on wheeze or objective measures of lower respiratory tract infection. Only a small proportion of episodes sought medical care (7%) or were absent from school or work (13%). Rates of medically attended illness were highest in infants, in keeping with the well-

described increased vulnerability to severe illness in this age group. RSV is estimated to cause >260,000 annual episodes of mild respiratory illness among children aged <5 years in South Africa, of which approximately 60,000 are medically attended[15]. RSV is also responsible for substantial cost burden in South Africa, accounting for 137,204,393 USD each year among children aged <5 years[16]. It is possible that frequent household visits may have affected health-seeking, biasing estimates of medically attended illness burden down.

Quantification of shedding duration and generation interval are important parameters for future models of RSV community transmission. We found that the mean duration of RSV shedding was 6.5 days, the same as 6.5 days in a study of influenza of similar design and within the range of previous studies[7,12,17–20]. We found the longest shedding in symptomatic individuals, and those with lower Ct values, as in our previous study of influenza and a Kenyan study of a similar design[12,17]. Similar to the Kenyan study, we identified a small number of individuals (mainly young children) who remained PCR positive for >21 days, potentially representing a reservoir of infection. A strength of our study is the systematic sampling irrespective of symptoms and testing with a sensitive rRT-PCR. RSV detection on rRT-PCR could, however, represent shedding of non-viable virus. Virus culture would be required to confirm the duration that these long-shedding individuals remain infectious to others, but was not possible in this study as samples were transported in Primestore MTM which inactivates viral pathogens while stabilizing RNA. The generation interval for RSV was 8.4 days and differed by RSV subgroup, longer than 5.9 days estimated for influenza in a similar study[17].

The overall HCIR was 11%, similar to that observed for influenza in the same population (10%) but substantially lower than for SARS-CoV-2 (23%) in the same community at a time when the population was largely naïve to SARS-CoV-2[17,21]. Importantly, children aged 1-12 years old accounted for 56% of index cases. Unexpectedly, of 12 infants who experienced ≥1 RSV infection episode, 10 were the index case at least once. The infection could have potentially been acquired from

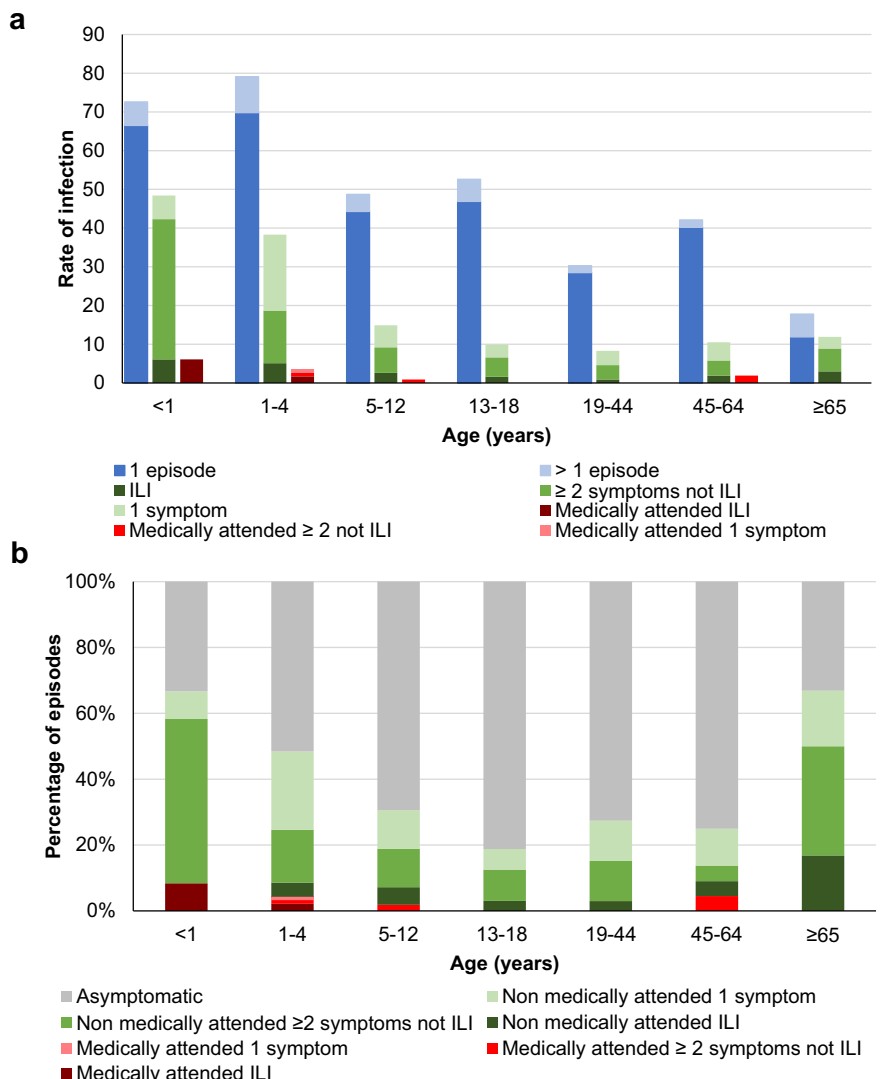

**Fig. 4 | Rates and symptomatic percentage of RSV infection and illness by age group at a rural and an urban site, South Africa, 2017-2018. a** Rates of RSV infections and RSV-associated illness per 100 person-years and **b** percentage of episodes by symptom and medical attendance. ILI – influenza-like illness (fever and cough).

caregivers outside the home or young children in neighbouring households, however, we did not collect data on this, limiting our ability to comment on the source of infections to infants. In addition, in some households not all household members participated in the study which could have led to misidentification of the index case in some instances. Previous studies have demonstrated an important role for toddlers and school-age children in RSV introduction. Munywoki et al. found that older children were the index case patients for 73% of within-household infant infections, and Hall et al. found that older siblings introduced infection in 50% of instances[5,7]. Other less intensive studies have identified siblings and mothers as sources of RSV infection to infants[22–25] Transmission was greater from symptomatic individuals and individuals who shed RSV for longer similar to Munywoki et al. and a study of influenza in the same cohort[5,17]. Individuals aged 1-4 and ≥65 years were more likely to acquire infection, possibly because of increased vulnerability related to less robust immunity[13]. Even though transmission was lower from asymptomatic individuals, it did occur at rates of 8%. Coupled with high rates of asymptomatic infection, asymptomatic individuals are likely important drivers of community transmission. Two-thirds of RSV infection episodes were presumed acquired in the community, similar to the estimates of 68% in a study of influenza in the same cohort[17].

Our study had limitations. Sampling for RSV every 3-4 days may have missed some infections of short duration and we had missing RSV rRT-PCR data for 10% of follow-up visits. Missed swabs were treated as negative which could have resulted in underestimation of the duration of infection or HCIR. RSV circulates year-round and we only followed up participants for 10 months of the year, while this period included the months of peak transmission, some cases may have been missed. Additionally, a small number of participants were enroled after the peak in RSV circulation. Our study was conducted at rural and urban sites in South Africa approximately 600 kilometres apart, which may not be representative of other settings; however, the similar estimates at both sites and over two years despite different climate and population characteristics suggests that this finding may be representative, at least for South Africa. Less than half of the approached households agreed to participate in our study, and we excluded households with <3 members, which could have introduced bias if included households differed from non-included households. Numbers for some subgroup analyses were small, leading to wide confidence intervals. Underlying illness was assessed by patient response, leading to possible under-reporting, and we did not collect data on childcare outside of the home.

Our study also had several strengths, including the inclusion of participants from two different communities who were followed

**Table 1 | Factors associated with symptomatic illness among RSV-infected individuals at a rural and an urban site, South Africa, 2017-2018[a]**

| Variable | | Symptomatic illness n/N (%) | Univariate OR[b] (95% CI) | Multivariable aOR[b] (95% CI) |
|---|---|---|---|---|
| Age group (years) | <1 | 8/12 (67) | 12.3 (2.4–62.0) | 10.8 (1.9–61.3) |
| | 1–4 | 45/93 (48) | 5.6 (2.3–13.4) | 5.3 (2.0–13.6) |
| | 5–12 | 34/111 (31) | 2.4 (1.1–5.6) | 2.5 (1.0–6.3) |
| | 13–18 | 12/64 (19) | Reference | Reference |
| | 19–44 | 18/66 (27) | 1.8 (0.7–4.6) | 2.2 (0.8–6.1) |
| | 45–64 | 11/44 (25) | 1.4 (0.5–4.1) | 2.0 (0.6–6.2) |
| | ≥65 | 4/6 (67) | 8.6 (1.1–70.8) | 12.6 (1.4–116.0) |
| Sex | Female | 81/236 (34) | 1.2 (0.7–1.9) | |
| | Male | 51/160 (32) | Reference | |
| HIV[c] | Infected | 11/48 (23) | 0.5 (0.2–1.1) | |
| | Uninfected | 114/336 (34) | Reference | |
| Other underlying illness | Absent | 125/384 (33) | Reference | |
| | Present | 7/12 (58) | 3.1 (0.8–12.4) | |
| Duration of shedding (days) | <4 | 27/125 (22) | Reference | Reference |
| | 4–10 | 74/211 (35) | 2.8 (1.6–4.9) | 1.8 (0.9–3.5) |
| | >10 | 31/60 (52) | 5.0 (2.2–11.2) | 3.1 (1.2–8.0) |
| Minimum Ct value | <30 | 103/240 (43) | 4.0 (2.3–7.1) | 2.5 (1.3–4.9) |
| | 30–37 | 29/156 (19) | Reference | Reference |
| Subgroup | A | 50/142 (35) | Reference | |
| | B | 79/229 (35) | 1.0 (0.6–1.7) | |
| | Mixed (A, B) | 1/4 (25) | 0.7 (0.1–9.1) | |
| | Untyped | 2/21 (10) | 0.1 (0.1–0.8) | |

[a] ≥ 1 symptom vs no symptoms reported among 396 RSV infection episodes (132, 33% with ≥1 symptom), four episodes with mixed subgroup infection each counted as a single episode.
[b] Odds ratios and p values estimated using mixed effects logistic regression adjusted for clustering by site and household.
[c] HIV status data available for 384/396 (97%) of individuals.
OR Odds ratio, aOR adjusted OR, CI confidence interval, Ct cycle threshold, n number. Additional factors evaluated but not found to be statistically significant include year, site, employment, education level, alcohol, smoking, cotinine level, underlying tuberculosis, receipt of influenza vaccine, body mass index.

intensively for 10 months, the inclusion of data for two consecutive seasons with high follow-up rates, and frequent sampling by rRT-PCR, irrespective of symptoms and systematic symptom ascertainment allowing for robust estimation of incidence and of the role of asymptomatic infections in transmission.

In conclusion, we describe high attack rates of RSV infections in rural and urban South African communities, with the highest rates in young children, who were also more likely to introduce infection into the home. These findings support further evaluation of whether upcoming RSV vaccines administered to toddlers or primary school children can prevent transmission to vulnerable individuals.

## Methods

### Study design and participants
This prospective cohort study was conducted in a rural and an urban community in South Africa from 2017 through 2018. The rural site in Mpumalanga Province is nested within a health and socio-demographic surveillance system (HDSS)[26,27]. The urban site is located in the North West Province (Supplementary Fig. 1).

### Ethics
The protocol was approved by the University of Witwatersrand Human Research Ethics Committee (Reference 150808). The protocol was registered on clinicaltrials.gov on 6 August 2015 (https://clinicaltrials.

gov/ct2/show/NCT02519803). The U.S. Centers for Disease Control and Prevention's Institutional Review Board relied on the local review (#6840). All participants or their caregivers provided written informed consent. In addition, participants received grocery store vouchers of USD 2-2.5 per visit to compensate for the discomfort and time associated with study procedures.

### Household selection
Household were randomly selected at each site. Descriptions of study sites and details of sampling are included in the supplement and have been published previously[17,28]. All members of selected households were approached for consent. Households with >2 members and where ≥80% of members individually consented to participate were enroled. Each year, we enroled new households, consecutively approached according to the sampling frame, until the sample size was reached.

### Data collection
We collected individual baseline data, including demographics and history of underlying illness. Cohort participants were followed up twice-weekly (Monday-Wednesday and Thursday-Saturday) from January-October through the RSV season. At each visit, irrespective of symptoms, nasopharyngeal swabs were collected and a questionnaire on symptoms, absenteeism, and health-seeking was completed. Field workers were trained in the identification of respiratory signs and symptoms. Data were entered during visits on tablet computers with use of the Research Electronic Data Capture application (REDCap).

### Specimen collection and laboratory testing
Nasopharyngeal samples were collected using nasopharyngeal nylon flocked swabs (PrimeSwab™, Longhorn Vaccines & Diagnostics, San Antonio, USA), placed in PrimeStore® Molecular Transport Medium (MTM) (Longhorn Vaccines & Diagnostics, San Antonio, USA) and transported on ice packs with temperature monitoring, to the National Institute for Communicable Diseases (NICD) in Johannesburg for testing. Nucleic acids were extracted using the Roche MagNA Pure 96 (Roche, Mannheim, Germany) according to the manufacturer's instructions. Nasopharyngeal samples were tested for RSV by real-time reverse transcription polymerase chain reaction (rRT-PCR) using the FTD Flu/RSV detection assay (Fast Track Diagnostics, Luxembourg). RSV subgroups were determined by an in-house assay, RSV A[29] and RSV B[30], respectively, using SuperScript® III RT-One Step RT-PCR System with Platinum™ Taq DNA Polymerase (Invitrogen, Waltham, Massachusetts, USA).

### Sample size
For the main study, we aimed to enroll approximately 1500 individuals (approximately 500 individuals per year) over three consecutive influenza and RSV seasons to allow the estimation of 20% risk of infection and a 10% risk of illness with 95% confidence intervals (CIs) and 5% absolute precision. Assuming an average household size of five individuals and a loss to follow-up of 10%, we aimed to enroll approximately 55 households with >2 household members per site each year with at least 50% having at least one child aged <5 years in the house[28]. Reliable symptom data were only available for 2017-2018, hence data from these years are included in the current analysis[17,28]. We performed an updated power calculation based on observed attack rates as well as the observed design effect of 1.8 (intra cluster correlation coefficient for the attack rate of RSV in households was 0.2). Based on these calculations we have 100% power to estimate a 43% attack rate of RSV.

### Definitions and analyses
Episodes and clusters of RSV infection were estimated separately by virus subgroup (Supplementary Fig. 2). Visits where there was no swab

**Table 2 | Factors associated with duration of RSV shedding at a rural and an urban site, South Africa, 2017-2018[a]**

| Variable | | Shedding duration (days) Mean (SD; range) | Univariate HR | Multivariable aHR |
|---|---|---|---|---|
| Age group (years) | <1 | 11.3 (11.7; 1–36) | 0.3 (0.1–0.7) | 0.5 (0.2–1.1) |
| | 1–4 | 7.4 (5.9; <1–32) | 0.6 (0.4–1.0) | 0.9 (0.6–1.4) |
| | 5–12 | 6.6 (4.4; <1–23) | 0.7 (0.5–1.1) | 0.8 (0.5–1.2) |
| | 13–18 | 6.2 (4.5; <1–24) | 0.9 (0.6–1.4) | 0.9 (0.6–1.4) |
| | 19–44 | 5.5 (6.7; < 1–50) | 1.0 (0.7–1.6) | 1.0 (0.7–1.7) |
| | 45–64 | 5.1 (4.1; <1–16) | Reference | Reference |
| | ≥65 | 7.2 (9.3; <1–25) | 0.7 (0.2–1.9) | 0.7 (0.2–1.9) |
| Sex | Female | 6.8 (6.0; <1–50) | 0.9 (0.7–1.2) | |
| | Male | 6.2 (5.1; <1–30) | Reference | |
| HIV | Infected | 6.9 (8.9; <1–50) | 0.9 (0.7–1.4) | |
| | Uninfected | 6.4 (5.1; <1–36) | Reference | |
| | Unknown | 7.5 (5.9; <1–23) | Not estimated | |
| Other underlying illness[b] | Absent | 6.5 (5.6; <1–50) | Reference | |
| | Present | 8.8 (6.9; 2–25) | 0.7 (0.4–1.3) | |
| Symptoms | None | 5.6 (5.0; <1–50) | Reference | Reference |
| | 1 | 7.1 (5.1; <1–30) | 0.6 (0.4–0.9) | 0.8 (0.5–1.1) |
| | ≥2 | 9.2 (7.3; <1–36) | 0.4 (0.3–0.6) | 0.6 (0.4–0.8) |
| Minimum Ct value | <30 | 8.4 (6.2; <1–50) | 0.2 (0.2–0.3) | 0.3 (0.2–0.4) |
| | 30–37 | 3.6 (2.7; <1–19) | Reference | Reference |
| Subgroup | A | 6.9 (6.2; <1–50) | Reference | |
| | B | 6.6 (5.5; <1–36) | 1.1 (0.8–1.4) | |
| | Untyped | 2.8 (2.0; <1–6) | 3.2 (1.8–5.7) | |

[a]Estimated using Weibull accelerated failure time regression adjusted for clustering by site and household, includes 400 episodes of RSV infection, 4 mixed subgroup infection counted as two separate infection episodes each.

[b]Self-reported history of asthma, lung disease, heart disease, stroke, spinal cord injury, epilepsy, organ transplant, immunosuppressive therapy, organ transplantation, cancer, liver disease, renal disease or diabetes.

*SD* Standard deviation, *HR* Hazard ratio, *Ct* cycle threshold.

Hazard ratio <1 corresponds to prolonged duration of shedding. Mean shedding duration 6.5 days, standard deviation 5.7 days, range <1-50 days.

Additional variables evaluated but found not to be associated with duration of symptoms include year, site, cotinine level, smoking, alcohol use.

collected were treated as negative. Details of definitions as well as a table of outcomes and definitions are provided in the supplement (Supplementary Table 1).

We defined an RSV infection episode as at least one nasopharyngeal swab rRT-PCR positive (cycle threshold (Ct) value <37) for RSV. We considered a new infection when the individual tested positive for a different subgroup or the same subgroup >2 weeks from the last day of the last previous positive; else, we considered it the same episode. This is because individuals could test negative and then positive again subsequently due to fluctuations of viral load or specimen quality. Episode duration in days was estimated from the first to the last day of rRT-PCR positivity plus a random number from a uniform distribution from 0-3 to account for the midpoint time from subsequent visits. An illness episode was defined as an episode with ≥1 symptom reported from the visit before to the visit after the RSV infection episode. Symptoms included: fever (self-reported or measured tympanic temperature ≥38∘C), cough, difficulty breathing, sore throat, nasal congestion, chest pain, muscle aches, headache, vomiting, or diarrhoea. Influenza-like illness (ILI) was defined as fever and cough within an RSV-confirmed episode. Medically attended illness was defined as an illness episode where the participant sought care with a nurse or physician during the episode. A lower Ct value (<30) on rRT-PCR was used as a proxy for a higher viral load.

A cluster was composed of all infections of the same subgroup within a household within an interval between infections of ≤2 mean serial intervals (3.5 days), including single infections. The household cumulative infection risk (HCIR) was defined as the cumulative number of all household members with RSV infection detected within a household cluster, divided by the total number of individuals participating in the study in the affected household, exclusive of the index case, restricted to secondary cases with first RSV positive <17 days after the index case first positive. The index case was defined as the first individual testing positive within a cluster. Households with co-primary index cases were excluded from the analysis of HCIR and the percent of infections acquired in the community. Using these definitions, it was possible for a household to experience >1 cluster of infection by the same subgroup or a different subgroup in the same season.

We defined the incidence of RSV infection or illness as the number of episodes divided by the person time under observation reported per 100 person years.

Proportions were compared using the Chi-squared or Fisher's exact test if the numbers were small. Factors associated with incidence were assessed using Poisson regression, factors associated with symptomatic illness and HCIR were estimated using logistic regression, and factors associated with shedding duration and generation interval were estimated using Weibull accelerated failure time regression. For the main analysis of incidence, we considered all identified episodes of infections, including >1 infection episode in the same individual within the same season. In addition, we performed an analysis considering at least one episode per season (excluding multiple infections). For all analyses, we accounted for within-household clustering using random effects regression models. For each univariate analysis, we used all available case information. For each multivariable model, we considered all a priori defined biologically associated factors with the outcome of interest for which

**Table 3 | Factors associated with household cumulative infection risk (HCIR)[a] at a rural and an urban site, South Africa, 2017-2018**

| Variable | | HCIR n/N (%) | Univariate OR[b] | Multivariable aOR[b] |
|---|---|---|---|---|
| **Characteristics of the index case** | | | | |
| Age group (years) | <1 | 5/22 (23) | 3.7 (0.5-28.7) | 1.0 (0.1–9.8) |
| | 1–4 | 26/203 (13) | 2.2 (0.6–8.4) | 1.1 (0.3–4.5) |
| | 5–12 | 40/267 (15) | 2.9 (0.8–10.1) | 1.1 (0.3–4.3) |
| | 13–18 | 11/151 (7) | 1.0 (0.2–4.5) | 0.6 (0.1–2.8) |
| | 19–44 | 9/116 (8) | 1.1 (0.3–5.1) | 1.0 (0.2–5.7) |
| | 45–64 | 5/93 (5) | Reference | Reference |
| | ≥65 | 0/4 (0) | Undefined | |
| Sex | Female | 49/391 (13) | 0.7 (0.3–1.3) | |
| | Male | 47/465 (10) | Reference | |
| HIV | Infected | 9/105 (9) | 0.8 (0.3–2.1) | |
| | Uninfected | 86/727 (12) | Reference | |
| Number of symptoms | 0 | 47/561 (8) | Reference | Reference |
| | 1 | 9/102 (8) | 1.0 (0.3–3.0) | 0.8 (0.3–2.6) |
| | ≥2 | 40/193 (21) | 4.0 (1.9–8.3) | 2.5 (1.1–5.3) |
| Duration of shedding (days) | <4 | 26/482 (5) | Reference | Reference |
| | 4–10 | 47/298 (16) | 5.0 (2.4–10.4) | 1.6 (20.5–5.8) |
| | >10 | 23/76 (30) | 11.6 (4.3–31.2) | 6.1 (1.5–26.2) |
| Minimum Ct value | <30 | 73/473 (15) | 4.9 (2.3–10.5) | |
| | 30–37 | 23/383 (6) | Reference | |
| **Characteristics of the household contact** | | | | |
| Age group (years) | <1 | 1/14 (7) | 1.7 (0.1–26.8) | 5.0 (0.3–99.5) |
| | 1–4 | 21/109 (19) | 5.4 (1.1–28.5) | 7.1 (1.2–41.5) |
| | 5–12 | 28/251 (11) | 2.1 (0.4–10.7) | 3.5 (0.6–19.0) |
| | 13–18 | 19/139 (14) | 2.6 (0.5–13.7) | 2.9 (0.5–16.3) |
| | 19–44 | 27/263 (10) | 1.9 (0.4–9.4) | 2.6 (0.5–13.9) |
| | 45–64 | 16/112 (14) | 2.1 (0.4–11.6) | 2.7 (0.5–15.7) |
| | ≥65 | 2/45 (4) | Reference | Reference |
| Sex | Female | 77/579 (13) | 1.3 (0.8–2.1) | |
| | Male | 37/354 (10) | Reference | |
| HIV | Infected | 11/136 (8) | 0.4 (0.2–0.9) | |
| | Uninfected | 99/764 (13) | Reference | |
| Other underlying illness | Absent | 112/905 (12) | Reference | |
| | Present | 2/28 (7) | 0.4 (0.1–2.2) | |

[a]The cumulative number of all household members with RSV infection detected within a household cluster, divided by the total number of individuals participating in the study in the affected household, exclusive of the index case, restricted to secondary cases with first RSV positive <17 days after the index case first positive. Clusters with different RSV subgroups which occur in the same time period counted separately.
[b]Odds ratios and p values estimated using logistic regression adjusted for clustering by site and household.
*CI* confidence interval, *Ct* cycle threshold.
Additional factors evaluated but not found to be statistically significant include year, site, employment of index or contact, education level of index or contact, alcohol or smoking of index or contact, urine cotinine level of index or contact, underlying tuberculosis, other underlying illness of index, body mass index of index or household contact, number of people in household, number of rooms, crowding, smoking inside the house, mean indoor summer and winter temperature, mean indoor summer and winter particulate matter, RSV subgroup.

we had available data. Age was included in all models as a possible confounder. As we examined risk factors associated with a number of different outcomes, selected predictors varied across models. Once we had developed the final models, we implemented a final model check using forward and backward selection using a two sided p value cut off of 0.05. Details of the multivariable model building process are shown in the supplement. The reference group for multi-level variables was chosen as the group with the lowest prevalence of the outcome of interest with sufficient numbers, and for time-to-event, the shortest duration. Pairwise interactions were assessed graphically and by inclusion of product terms for all variables remaining in the final multivariable additive model. We conducted all statistical analyses using Stata version 14.1 (Stata Corp LP, College Station, Texas, USA). Sensitivity analyses are described in the supplement. The full code is available on zenodo[31].

### Study protocol

The full study protocol can be found on the National Institute for Communicable Diseases Website at the following link: https://crdm.nicd.ac.za/projects/phirst/

### Data availability

All data associated with this study are present in the paper or the Supplementary Information. Individual-level data cannot be publicly shared because of ethical restrictions and the potential for identifying included individuals. Accessing individual participant data and a data

dictionary defining each field in the dataset would require provision of protocol and ethics approval for the proposed use. To request individual participant data access, please submit a proposal to C.C. who will respond within 1 month of request. Upon approval, data can be made available through a data sharing agreement. Aggregate data to reproduce the figures are available at https://doi.org/10.5281/zenodo.10117300.

## Code availability

Analysis code available at https://doi.org/10.5281/zenodo.10117300.

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

## Acknowledgements

The authors would like to thank all the individuals who kindly agreed to participate in the study as well as the PHIRST group. This work was supported by the U.S. Centers for Disease Control and Prevention [cooperative agreement number: 5U51IP000155] (CC). Testing for RSV was supported by the Bill and Melinda Gates Foundation (Grant number: OPP1164778) (NW). The MRC/Wits Rural Public Health and Health Transitions Research Unit and Agincourt Health and Socio-Demographic Surveillance System, a node of the South African Population Research Infrastructure Network (SAPRIN), is supported by the Department of Science and Innovation, the University of the Witwatersrand, and the Medical Research Council, South Africa, and previously the Wellcome Trust, UK (grants 058893/Z/99/A; 069683/Z/02/Z; 085477/Z/08/Z; 085477/B/08/Z). The findings and conclusions in this report are those of the author(s) and do not necessarily represent the official position of their institutions and/or funding agencies.

## Author contributions

Conception and design of the study: C.C., J.M., M.L.M., F.K.T., O.H., A.V.G., N.W., S.T. Data collection and laboratory processing: C.C., J.K., J.M., M.L.M., F.K.T., O.H., Az.M., M.C., A.V.G., N.W., N.A.M., K.K., L.L., K.M., F.W, F.X.G., T.M., An.M., S.P., B.L., S.T. Analysis and interpretation: C.C., J.K., J.M., M.L.M., F.K.T., O.H., Az.M., M.C., A.V.G., N.W., N.A.M., K.K., L.L., K.M., F.W., F.X.G., T.M., An.M., S.P., B.L., S.T. Drafting or critical review of the article: C.C., J.K., J.M., M.L.M., F.K.T., O.H., Az.M., M.C., A.V.G., N.W., N.A.M., K.K., L.L., K.M., F.W., F.X.G., T.M., An.M., S.P., B.L., S.T.

## Competing interests

C.C. has received grant support from Sanofi Pasteur, US CDC, Welcome Trust, Programme for Applied Technologies in Health (PATH), Bill & Melinda Gates Foundation and South African Medical Research Council (SA-MRC). A.V.G has received grant support from CDC, ASLM/Africa CDC, SA-MRC, WHO Afro, Fleming Fund, WHO, Wellcome Trust. N.W. reports receiving grants from Sanofi Pasteur, US CDC and the Bill & Melinda Gates Foundation. N.A.M has received a grant to his institution from Pfizer to conduct research in patients with pneumonia. J.M. has received grant support from Sanofi Pasteur and PATH. M.C received the Robert Austrian Award sponsored by Pfizer as well as received funding as part of the South Africa-Pittsburgh Public Health Genomic Epidemiology (SAPPHGenE) training program and reports support for attending meetings and/or travel paid to the institution from Bill and Melinda Gates Foundation. Other authors declare no competing interests.

## Additional information

[1]Centre for Respiratory Diseases and Meningitis, National Institute for Communicable Diseases of the National Health Laboratory Service, Johannesburg, South Africa. [2]School of Public Health, Faculty of Health Sciences, University of the Witwatersrand, Johannesburg, South Africa. [3]Coronavirus and Other Respiratory Viruses Division (proposed), Centers for Disease Control and Prevention, Atlanta, GA, USA. [4]Influenza Program, Centers for Disease Control and Prevention, Pretoria, South Africa. [5]School of Pathology, Faculty of Health Sciences, University of the Witwatersrand, Johannesburg, South Africa. [6]Perinatal HIV Research Unit, MRC Soweto Matlosana Collaborating Centre for HIV/AIDS and TB, University of the Witwatersrand, Johannesburg, South Africa. [7]DST/NRF Centre of Excellence for Biomedical Tuberculosis Research, University of the Witwatersrand, Johannesburg, South Africa. [8]Johns Hopkins University Center for TB Research, Baltimore, MD, USA. [9]MRC/Wits Rural Public Health and Health Transitions Research Unit (Agincourt), Faculty of Health Sciences, School of Public Health, University of the Witwatersrand, Johannesburg, South Africa. [10]Environment and Health Research Unit, South African Medical Research Council, Johannesburg, South Africa. [11]Unit for Environmental Science and Management, Climatology Research Group, North-West University, Potchefstroom, South Africa. [12]MassGenics, Duluth, GA, USA. ✉e-mail: cherylc@nicd.ac.za

