## [Peer Review File · Nature Communications]

Incidence and transmission of respiratory syncytial virus in urban and rural South Africa, 2017-2018REVIEWER COMMENTS

Reviewer #1 (Remarks to the Author):

With interest, I have read the manuscript of Cohen et al. titled "Incidence and transmission of respiratory syncytial virus in urban and rural communities in South Africa, 2017-2018: results of the PHIRST cohort study". This large observational prospective household study on RSV incidence and household transmission was conducted in South Africa during two consecutive seasons at one urban and one rural site. The main strength of this study is the systematic sampling of all participating household members, regardless of the presence of respiratory symptoms and its outstanding retainment rate, despite a very intensive study design. The authors found a high incidence of RSV infections, particularly in young children. In most RSV family clusters, older siblings were responsible for introducing the virus into the household. Remarkably, in 10 out of 12 RSV infection episodes in infants, the infant itself was the index case. Subsequently to the introduction of RSV in the house, 11% of household members other than the index case were infected with RSV, with children aged 1-4 years most likely to be infected. Overall, this is a well-conceived and very complete study on RSV transmission in the community. Household transmission studies are necessary to assess if childhood vaccines against RSV that are currently under development could reduce RSV transmission and this study will contribute to this knowledge. However, the article lacks focus and although the authors claim that the results point to a high potential of childhood RSV vaccines, older siblings were not the main source of infection for infants in this study.

Major comments

The article lacks focus which makes it difficult to read/understand. The results section contains a lot of information without any indication what outcomes the authors deem important. Could the authors indicate, e.g. by adding headings to each paragraph, the most important outcomes.

How did the authors account for the fact that not all household members had to participate in the study when analyzing household transmission? How could the index case be

ascertained? Information about how many household members were not included is missing. In addition, 60 individuals were excluded from analysis.

It would be helpful to add the number of households in which all household members participated and were included in the analysis and analyze whether the results about transmission are similar compared to the analysis of all households. I would suggest to also add this as a limitation to the discussion.

Methods and supplementary methods

There is a considerable overlap between methods and supplementary methods. The methods section is now quite long, which makes it difficult to determine the focus of the manuscript. Could the authors be more selective in the information they give in the methods and add less relevant information to the supplementary methods without duplicating information. E.g. line 128-137 can be added to the suppl methods.

There are several inconsistencies in outcomes definitions between the supplementary materials and the article. Could the authors please review and address these?

- In the supplementary data the sample size calculations are for SARS-CoV-2 for a study conducted in 2017-2018!
- The HCIR is defined in table 3 caption: as the proportion of household members infected in a cluster divided by the number of household members participating in the study. But is referred to as a proportion (number of household members infected?)
- Line 169-180 a cluster is defined as “all infections of the same subgroup within a household within an interval between infections of ≤ 2 mean serial intervals (3.5 days), including single infections” and the HCIR as “the cumulative proportion of all household members with RSV infection detected within a household cluster, divided by the total number of individuals participating in the study in the affected household”. However, it is mentioned on L185 “We included all secondary cases who were PCR positive <17 days after the index case in analyses of generation interval and HCIR.”
- The definition of an RSV episode

Line 161: “Episode duration in days was estimated from the first to the last day of rRT-PCR positivity plus three”

Suppl Fig 2.: “Episode duration was estimated from the first to the last day of RSV rRT-PCR positivity with the same subgroup”

Minor comments

Abstract/Introduction

It is not immediately clear that this study comprised 2 seasons, please consider to add this for clarity.

Is South Africa likely to be representative of Sub-Saharan Africa, as suggested in the abstract?

Line 75 ...European Medicines Agency in 2022 or and others anticipated in 2023.

This sentence is not clear, please adjust

Methods

Line 111 and 649 In the rural site, households were randomly selected...

Could the authors add some more explanation how this was done in the supplementary files.

Results

Line 216 Reasons for exclusion from analysis for 60 individuals included refusal, withdrawal of consent or relocation.

Suppl figure 3: the following reasons for exclusion were relocation, ≤ 10 swabs collected and death.

Could the authors clarify this inconsistency in exclusion reasons?

Line 240 There were 396 infection episodes in total, four of which had a mixture of RSV

subtypes.

Consider to move the last part to the paragraph about type A and B.

Line 243 §15% (n=37) 13-18 years

§ is a typing error?

Line 254-255

...with a higher proportion of symptomatic infections in individuals aged <5 and ≥65 years

Please add proportion

Line 264

...3% (12/400) of episodes shed for >21 days (3 aged <1, 4 aged 1-4 and 2 aged 5-12 years).

What was the age of the other 3 episodes?

Line 275-281

The overall HCIR was 11% (96/856) and 61% (147/242) of clusters included only one individual. Transmission was highest from index cases with ≥2 symptoms (21%, 40/193) vs. asymptomatic individuals (8%, 47/561) (Table 3). On multivariable analysis controlling for index case age, factors associated with increased transmission were ≥2 symptoms vs. no symptoms and duration of shedding 4-10 or >10 days vs <4 days. Individuals aged 1-4 years vs. ≥65 years were more likely to acquire RSV infection. Among 394 infection episodes in RSV clusters without coprimary index cases, 264 (67%) infections were presumed acquired in the community (i.e., were the index case).

It is not clear where the denominator numbers come from. Please add some more explanation.

Two-third of infection episodes are acquired in the community, so it seems transmission rate within the household is limited. Could this also be partly due to missing data from

household members who did not participate or were excluded from analysis? Is this percentage similar in households with complete participation of all household members? See also major comment.

Line 308 We did not find any association with HIV infection status and incidence of infection, symptomatic fraction, or acquisition of infection.

Could this also be due to a low sample size for this subgroup?

Discussion

The authors underlined having two sites in urban/rural settings as an important strength, yet almost all outcomes are reported on pooled analysis. The authors report similar results. Could it be possible that a combination of different factors resulted in similar incidence despite differences in circulating RSV type and baseline characteristics?

Line 554 HIV status data available for 374 /396 (94%) of individuals.

If infected and uninfected in table are added it is 384 instead of 374 (48+336).

Line 586 Please add a label for Figure 1

Supplementary files

I would like to respectfully suggest reorganizing the methods section of the supplementary data to improve readability, for example, to provide outcome definitions as a list or table.

Line 623-632

Based on the wording in the supplementary material, the sample size seemed to have been calculated for estimating RSV incidence as attack rates in each age group at individual level. Could the authors clarify the implication for the study power considering incidence was reported per 100 person-years and how the 1.3 value for the design effect was estimated.

Line 729 likely because the viral load was too low...

Please correct

Line 877 Suppl Table 1: Baseline data. The p-values are abnormally small for the number of participants and the difference between groups for the parts about individual level characteristics (e.g. children <1: n=9 (2%) vs n=13 (2%), p-value=0.03!). Could the authors provide an explanation how these p-values were calculated?

Reviewer #2 (Remarks to the Author):

This study outlines an analysis of new data taken from a previously described prospective cohort study on the prevalence and incidence of RSV in an LMIC setting. It is one of the largest and most comprehensive studies of community-acquired RSV infection and disease conducted in an LMIC setting. The results are particularly timely and interesting to policymakers given the huge interest in RSV. They will be important in determining the impact and cost-effectiveness of upcoming prophylactics against disease in these settings. Therefore, this important piece of work should be published. The methods seem sound but are vague and lack clarity in places; I have a few minor comments which might help this. There are also a few typos and errors in some of the tables and figures I've highlighted a few below. I would like to see this addressed before publication. Providing these are responded to appropriately; this paper should be published in Nature Communications.

Minor comments

Manuscript:

To account for left and right censoring a period of 3 days is added to all specimens to estimate the missed duration of infection. To properly reflect the uncertainty in the duration of infection, a uniform distribution of between 0–3 days at each end should be included. Also, the cluster duration and generation interval time estimate ignores left/right censoring from the biweekly sampling. Is there a reason for this? How will this change the

interpretation of the results?

Could you specify in the methods how you dealt with missed infections (about 10% of samples were missed) and how they were used in calculating the duration of infection/cluster duration/generation interval calculation?

Could you provide a bit more information about the statistical methods used? Particularly the Weibull accelerated failure time model; I understand this is an alternative to a proportional hazard model, but unclear why this was chosen. Just a few equations in the SI would help clarify. Also, lines 199–201 are really vague, how were selected predictors varied across models? Also, the forwards backwards selection regression algorithms aren't always the most reliable, could you provide some outputs for which variables were removed at each step?

The authors should also test how sensitive these results are to their definition of a new infection being constrained to >2 weeks from the last day of the previous infection. Given shedding is very long for some individuals, a sensitivity analysis with 1> and >3 weeks would shed light on this.

Figures:

Though Figure 1 is great for getting an overview of the epidemiology, it would be of particular interest to modellers if you could plot histograms of the duration of shedding for significant covariates (i.e. age, and symptoms). Particularly given the large positive skew (e.g. how often are people shedding for longer than 14 days? Seems there are a fair few in Table 2 but these might just be a handful of super-long shedders!)

Figure 1, the y-axis is "cumulative attack rate" no?

There also appear to be some people recruited after the peak of incidence, (Figure 1b for example) have the incidence rates been adjusted to account for this?

Figure 3a, could you add some grid lines?

Tables in Supp:

Table S1: It's unclear what the difference between the number of rooms for sleeping (1–2[which group is 2 in?], 2–4, >4) and the number of rooms for sleeping (itself) is.

It's also unclear what the reference is for some of the pK values, (e.g. crowding, children aged >5 years in the household, Female sex etc.) There are a few typos in the captions of this table too.

Table S5: What is A/B? (Perhaps this is in the manuscript, apologies if I missed this.)

Table S6 (2020/21 in the caption)? vs Table S7? I'm not quite sure what the difference between these Tables is, could you spell this out a bit more?

Reviewer #3 (Remarks to the Author):

This is a groundbreaking study. The authors have done a phenomenal job in conducting a prospective household follow-up study of such scale and intensity and report some remarkable findings on the local transmission of RSV in South Africa. Although a study of a similar design was conducted in Kenya over a decade ago, the size and sampling intensity conducted here is truly commendable. The results of the study provide a strong evidence base upon which local policy decisions relating to RSV vaccination can be made.

Minor points

1. The authors report that the rate of medically-attended infection in infants at 6.1 per 100 persons. Are there similar estimates for older adults over the age of 65? Secondly, because the rates of medically attended infection do not adequately reflect the burden of disease in LMIC settings, did the study collect any objective measures of lower respiratory tract infection in infants and older adults (difficulty in breathing, lower chest wall indrawing, fingertip pulse oximetry or lung function?)

2. What are the seasonality patterns of RSV in Mpumalanga Province? Was the follow-up synchronized to match seasonal transmission?

3. I was struck by the estimated incidence of illness which was greater in 5-12 years olds (14.9/100pyo) compared to older adults >65y (11.9/100py). Could the authors comment?

4. At least one participant was reported to have shed virus for 53 days. Shedding duration was reportedly associated with age, symptoms and viral load. Could this also potentially be explained by HIV status

REVIEWER COMMENTS

Reviewer #1 (Remarks to the Author):

With interest, I have read the manuscript of Cohen et al. titled “Incidence and transmission of respiratory syncytial virus in urban and rural communities in South Africa, 2017-2018: results of the PHIRST cohort study” . This large observational prospective household study on RSV incidence and household transmission was conducted in South Africa during two consecutive seasons at one urban and one rural site. The main strength of this study is the systematic sampling of all participating household members, regardless of the presence of respiratory symptoms and its outstanding retention rate, despite a very intensive study design. The authors found a high incidence of RSV infections, particularly in young children. In most RSV family clusters, older siblings were responsible for introducing the virus into the household. Remarkably, in 10 out of 12 RSV infection episodes in infants, the infant itself was the index case. Subsequently to the introduction of RSV in the house, 11% of household members other than the index case were infected with RSV, with children aged 1-4 years most likely to be infected. Overall, this is a well-conceived and very complete study on RSV transmission in the community. Household transmission studies are necessary to assess if childhood vaccines against RSV that are currently under development could reduce RSV transmission and this study will contribute to this knowledge. However, the article lacks focus and although the authors claim that the results point to a high potential of childhood RSV vaccines, older siblings were not the main source of infection for infants in this study.

We thank the reviewer for these positive comments. Throughout and in response to the detailed comments below we have revised the paper for clarity and focus.

We agree that older siblings were not the main source of infection for infants and we have amended the abstract to reflect this and updated the discussion to more fully discuss these findings and possible implications.

Abstract page 3 lines 63-64: Future studies should examine whether vaccines targeting children aged <12 years could reduce community transmission.

Discussion page 15 line 410-414: Infection could have potentially been acquired from caregivers outside the home or young children in neighbouring households, however we did not collect data on this, limiting our ability to comment on the source of infections to infants. In addition, in some households not all household members participated in the study which could have led to misidentification of the index case in some instances.

Major comments

The article lacks focus which makes it difficult to read/understand. The results section contains a lot of information without any indication what outcomes the authors deem important. Could the authors indicate, e.g. by adding headings to each paragraph, the most important outcomes.

We have reviewed the paper extensively to improve readability and clarity. Headings have been added to each paragraph. Some results deemed to be less important have been moved to the supplementary results section to improve the flow and readability.

How did the authors account for the fact that not all household members had to participate in the study when analyzing household transmission? How could the index case be ascertained? Information about how many household members were not included is missing. In addition, 60 individuals were excluded from analysis.

It would be helpful to add the number of households in which all household members participated and were included in the analysis and analyze whether the results about transmission are similar compared to the analysis of all households. I would suggest to also add this as a limitation to the discussion.

As indicated in Figure 3 there were 1176 individuals in the 225 included households, of whom 60 (5%) were not included.

The text of results has been updated to clarify this page 10 lines 245-248: Of 1176 household members, 1116 (95%) were included in the analysis. Reasons for exclusion from analysis for the remaining 60 individuals included relocation, death or ≤ 10 swabs collected because of refusal, or withdrawal of consent or relocation.

The suggested sensitivity analysis restricted to households in which all household members participate has been performed and the following text has been added.

*Supplementary methods page 46 lines 901-904:
Sensitivity analyses*

In order to explore possible bias introduced by the fact that in some households, not all members participated in the study, we performed a sensitivity analysis including only households with all members participating.

*Supplementary results page 48 lines 925-934
Sensitivity analyses*

Of 225 included households, 151 households (with 717 household members) had all household members included in the study. On sensitivity analysis restricted to this subset of households, HCIR was similar to the main analysis (11%, 60/539). Factors associated with HCIR were similar to the main analysis in magnitude and direction, but the association with age of the household member was no longer statistically significant, likely as a result of low numbers (data not shown in supplement, results of this analysis, together with analysis code available at https://github.com/crdm-nicd/phirst_rsv). The proportion of infection episodes acquired in the community was similar on sensitivity analysis (68%, 154/228).

Discussion page 15, lines 412-414:

In addition, in some households not all household members participated in the study which could have led to misidentification of the index case in some instances.

Methods and supplementary methods

There is a considerable overlap between methods and supplementary methods. The methods section is now quite long, which makes it difficult to determine the focus of the manuscript. Could the authors be more selective in the information they give in the methods and add less relevant information to the supplementary methods without duplicating information. E.g. line 128-137 can be added to the suppl methods.

We have reviewed the methods section and reduced the length substantially, moving information to the supplement where needed. We have also reviewed for duplication and removed this. We have elected to retain lines 128-137 in the methods as these describe the primary sample collection, transport and testing methods. Because changes are quite substantial they are not all pasted here in response to reviewers but can be viewed in the attached tracked manuscript file.

There are several inconsistencies in outcomes definitions between the supplementary materials and the article. Could the authors please review and address these?

We thank the reviewer for identifying these errors which have been corrected. We have also added a table of outcome definitions (Supplementary table 1) on page 55.

- In the supplementary data the sample size calculations are for SARS-CoV-2 for a study conducted in 2017-2018!

The erroneous sample size information in the supplement has been deleted. Correct information is included in the methods section and expanded on response to queries from reviewer 2 detailed below.

- The HCIR is defined in table 3 caption: as the proportion of household members infected in a cluster divided by the number of household members participating in the study. But is referred to as a proportion (number of household members infected?)
- Line 169-180 a cluster is defined as “all infections of the same subgroup within a household within an interval between infections of ≤ 2 mean serial intervals (3.5 days), including single infections” and the HCIR as “the cumulative proportion of all household members with RSV infection detected within a household cluster, divided by the total number of individuals participating in the study in the affected household” . However, it is mentioned on L185 “We included all secondary cases who were PCR positive <17 days after the index case in analyses of generation interval and HCIR.”

We have reviewed the definitions of the calculation of HCIR and updated throughout the paper to ensure correctness and consistency. We have also added a table of definitions for the different outcomes to improve clarity (Supplementary table 1).

Methods page 7 lines 196-200: The household cumulative infection risk (HCIR) was defined as the cumulative number of all household members with RSV infection detected within a household cluster, divided by the total number of individuals participating in the study in the affected household, exclusive of the index case, restricted to secondary cases with first RSV positive <17 days after the index case first positive.

Footnote to table 3: “The cumulative number of all household members with RSV infection detected within a household cluster, divided by the total number of individuals participating in the study in the affected household, exclusive of the index case, restricted to secondary cases with first RSV positive <17 days after the index case first positive.

Supplementary methods page 45 lines 855-864: The generation interval was calculated as the difference between the dates of the first positive PCR tests in the index case and in the secondary case, each adjusted by adding a random number selected from a uniform

distribution between 0 and 3 (inclusive). For the analysis of generation interval and HCIR, we included all secondary cases with PCR positivity <17 days after the index case. This was chosen as the 75% quantile of the of the generation intervals observed in our cohort was 16 days (Figure 3). Infection episodes after this period are more likely to be tertiary or quaternary transmission events. As index case characteristics associated with transmission parameters was a focus of our analysis, we tried to limit the analysis to early transmissions in the household.

Supplementary table 1 page 55:

Household cumulative infection risk (HCIR) - The cumulative number of all household members with RSV infection detected within a household cluster, divided by the total number of individuals participating in the study in the affected household, exclusive of the index case. Restricted to clusters without coprimary index cases. Restricted to secondary cases with first RSV positive <17 days after the index case first positive. Included mixed subgroup infections as two separate infections.

- The definition of an RSV episode

Line 161: “Episode duration in days was estimated from the first to the last day of rRT-PCR positivity plus three”

Suppl Fig 2.: “Episode duration was estimated from the first to the last day of RSV rRT-PCR positivity with the same subgroup”

In response to suggestions from reviewer 2 we have updated the approach to estimation of episode duration to add a number selected from a random distribution from 0-3 to account for missed visits. Methods have been updated throughout to consistently reflect this approach. These are detailed in the response to reviewer 2 below.

We have also updated supplementary figure 2 to reflect the updated definitions.

Minor comments

Abstract/Introduction

It is not immediately clear that this study comprised 2 seasons, please consider to add this for clarity.

This has been added abstract page 3 lines 50-51: we conducted a prospective cohort study in rural and urban communities over two seasons during 2017-2018

Is South Africa likely to be representative of Sub-Saharan Africa, as suggested in the abstract?

This has been removed from the abstract page 3 line 49: Data on RSV incidence and household transmission are limited.

Line 75 ···European Medicines Agency in 2022 or and others anticipated in 2023.

This sentence is not clear, please adjust

This has been updated page 4 line 77: European Medicines Agency in 2022 and others anticipated in 2023³

Methods

Line 111 and 649 In the rural site, households were randomly selected...

Could the authors add some more explanation how this was done in the supplementary files.
This has been added, supplementary methods page 41 lines 733-736: Within these villages, households known to have >2 members from a census conducted in the previous year or more recently, were randomly selected. Simple random sampling was performed using a complete list of all households with >2 members within the selected villages.

Results

Line 216 Reasons for exclusion from analysis for 60 individuals included refusal, withdrawal of consent or relocation.

Suppl figure 3: the following reasons for exclusion were relocation, ≤ 10 swabs collected and death.

Could the authors clarify this inconsistency in exclusion reasons?

Text has been updated, results page 10 lines 245-248: Reasons for exclusion from analysis for the remaining 60 individuals included relocation, death or ≤ 10 swabs collected because of refusal or withdrawal of consent.

Line 240 There were 396 infection episodes in total, four of which had a mixture of RSV subtypes.

Consider to move the last part to the paragraph about type A and B.

Thanks for the suggestion we have elected not to move the sentence because where it is placed it follows on from the explanation of repeat infections allowing the reader who wishes to do so to calculate the total number of infections (356+31(two infection episodes) +3+3 (three infection episodes)=396) and relate this to the number of infected individuals specified in the same paragraph.

Line 243 § 15% (n=37) 13-18 years

§ is a typing error?

This does not appear in my version – perhaps an error in the file conversion

Line 254-255

...with a higher proportion of symptomatic infections in individuals aged <5 and ≥ 65 years

Please add proportion

This has been added results page 11 line 298: with a higher proportion of symptomatic infections in individuals aged <5 (50%) and ≥ 65 (67%) years

Line 264

...3% (12/400) of episodes shed for >21 days (3 aged <1 , 4 aged 1-4 and 2 aged 5-12 years).

What was the age of the other 3 episodes?

This has been added results page 12 lines 310-311: (3 aged <1 , 4 aged 1-4 and 2 aged 5-12, 2 aged 13-44 and 1 aged ≥ 65 years).

Line 275-281

The overall HCIR was 11% (96/856) and 61% (147/242) of clusters included only one individual. Transmission was highest from index cases with ≥ 2 symptoms (21%, 40/193) vs. asymptomatic individuals (8%, 47/561) (Table 3). On multivariable analysis controlling for index case age, factors associated with increased transmission were ≥ 2 symptoms vs. no symptoms and duration of shedding 4-10 or >10 days vs <4 days. Individuals aged 1-4 years vs. ≥ 65 years were more likely to acquire RSV infection. Among 394 infection episodes in RSV clusters without coprimary index cases, 264 (67%) infections were presumed acquired in the community (i.e., were the index case).

It is not clear where the denominator numbers come from. Please add some more explanation. Two-third of infection episodes are acquired in the community, so it seems transmission rate within the household is limited. Could this also be partly due to missing data from household members who did not participate or were excluded from analysis? Is this percentage similar in households with complete participation of all household members? See also major comment.

The denominators for the HCIR estimates are numbers of exposed household members. The proportion of clusters with one individual is out of all included clusters.

This is now clarified in the text results page 12 lines 322-325: The overall HCIR was 11% (96 of 856 exposed household members) and 61% (147) of 242 clusters included only one individual. Transmission was highest from index cases with ≥ 2 symptoms (21%, 40 of 193 exposed household members) vs. asymptomatic individuals (8%, 47 of 561 exposed household members) (Table 3).

The two thirds of infections acquired in the community is similar to for influenza. Text has been added to the discussion page 15 lines 424-425: Two thirds of RSV infection episodes were presumed acquired in the community, similar to the estimates of 68% a study of influenza in the same cohort¹⁰.

To explore the effect of missing data from household members who did not participate or were excluded from analysis we conducted the suggested sensitivity analysis.

This has been added to the supplement

Methods page 46 lines 902-904: In order to explore possible bias introduced by the fact that in some households, not all members participated in the study, we performed a sensitivity analysis restricted to households with all members participating.

Supplementary results page 48 line y932-933: The proportion of infection episodes acquired in the community was similar on sensitivity analysis (68%, 154/228).

Line 308 We did not find any association with HIV infection status and incidence of infection, symptomatic fraction, or acquisition of infection.

Could this also be due to a low sample size for this subgroup?

We agree and we have amended the discussion as follows as follows page 13 lines 363-366: An important limitation of this analysis was the relatively small numbers of PLWH included in the study which may have limited power to detect associations. We were also not able to compare the characteristics of PLWH well-controlled on treatment to those poorly controlled.

Discussion

The authors underlined having two sites in urban/rural settings as an important strength, yet almost all outcomes are reported on pooled analysis. The authors report similar results. Could it be possible that a combination of different factors resulted in similar incidence despite differences in circulating RSV type and baseline characteristics?

Incidence did vary by site and year. These data are presented in supplementary table 2. The results section has been amended to highlight this fact.

Results page 10 lines 264-265: The incidence estimates of RSV infection and illness (at least one symptom) were 47.2 and 15.7 per 100 person-years, respectively but varied by site and year

Line 554 HIV status data available for 374 /396 (94%) of individuals.

If infected and uninfected in table are added it is 384 instead of 374 (48+336).

We thank the reviewer for identifying this error and have updated the text to reflect the correct total (n=384).

Line 586 Please add a label for Figure 1

Figure 1 has been updated

Supplementary files

I would like to respectfully suggest reorganizing the methods section of the supplementary data to improve readability, for example, to provide outcome definitions as a list or table.

Thank you for this helpful suggestion. We have reorganised supplementary methods including adding subheadings and removing duplication with main methods. We have added the suggested table of definitions of outcome (Supplementary table 1).

Line 623-632

Based on the wording in the supplementary material, the sample size seemed to have been calculated for estimating RSV incidence as attack rates in each age group at individual level. Could the authors clarify the implication for the study power considering incidence was reported per 100 person-years and how the 1.3 value for the design effect was estimated.

The sample size explanation has been updated. We have also added an updated power calculation and included in this an updated estimate of the design effect using the observed data. We chose to report the incidence per 100 person-years for comparability with other studies but given the very good follow up the numbers are very similar whether expressed as an attack rate or an incidence per 100 person season hence this not affect the power calculations.

Methods page 6 lines 158-196:

For the main study, we aimed to enroll approximately 1500 individuals (approximately 500 individuals per year) over three consecutive influenza and RSV seasons to allow the estimation of 20% risk of infection and a 10% risk of illness with 95% confidence intervals (CIs) and 5% absolute precision. Assuming an average household size of five individuals and a loss to follow-up of 10%, we aimed to enroll approximately 55 households with >2 household members per site each year with at least 50% having at least one child aged <5 years in the house⁹. Reliable symptom data were only available for 2017-2018, hence data

from these years are included in the current analysis^{9,10}. We performed an updated power calculation based on observed attack rates as well as the observed design effect of 1.8 (intra cluster correlation coefficient for the attack rate of RSV in households was 0.2). Based on these calculations we have 100% power to estimate a 43% attack rate of RSV.

Line 729 likely because the viral load was too low...

Please correct

This has been corrected

Line 877 Suppl Table 1: Baseline data. The p-values are abnormally small for the number of participants and the difference between groups for the parts about individual level characteristics (e.g. children <1: n=9 (2%) vs n=13 (2%), p-value=0.03!). Could the authors provide an explanation how these p-values were calculated?

The analysis was done using logistic regression adjusted for clustering by site and household. Results have been verified as correct. A footnote in table (indicated by the superscript k) describes how the p values were calculated.

Reviewer #2 (Remarks to the Author):

This study outlines an analysis of new data taken from a previously described prospective cohort study on the prevalence and incidence of RSV in an LMIC setting. It is one of the largest and most comprehensive studies of community-acquired RSV infection and disease conducted in an LMIC setting. The results are particularly timely and interesting to policymakers given the huge interest in RSV. They will be important in determining the impact and cost-effectiveness of upcoming prophylactics against disease in these settings. Therefore, this important piece of work should be published. The methods seem sound but are vague and lack clarity in places; I have a few minor comments which might help this. There are also a few typos and errors in some of the tables and figures I've highlighted a few below. I would like to see this addressed before publication. Providing these are responded to appropriately; this paper should be published in Nature Communications.

We thank the reviewer for these positive comments and have responded individually to the detailed suggestions below.

Minor comments

Manuscript:

To account for left and right censoring a period of 3 days is added to all specimens to estimate the missed duration of infection. To properly reflect the uncertainty in the duration of infection, a uniform distribution of between 0-3 days at each end should be included. Also, the cluster duration and generation interval time estimate ignores left/right censoring from the biweekly sampling. Is there a reason for this? How will this change the interpretation of the results?

Thank you for this extremely helpful suggestion. We have implemented the suggested analysis to properly reflect the uncertainty for episode duration. We have also updated the approach to cluster duration and generation interval estimation to also account for this uncertainty.

The results remain very similar to those originally presented but we agree with the reviewer that their suggested approach is more robust.

Methods page 7 lines 182-184: Episode duration in days was estimated from the first to the last day of rRT-PCR positivity plus a random number from a uniform distribution from 0-3 to account for the midpoint time from subsequent visits.

Supplementary methods page 45 lines 855-857: The generation interval was calculated as the difference between the dates of the first positive PCR tests in the index case and in the secondary case, each adjusted by adding a random number selected from a uniform distribution between 0 and 3 (inclusive).

Supplementary table 1:

Cluster duration - The interval from the first day of positivity of the first individual in a cluster to the last day of positivity of the last individual. First and last day adjusted by adding a random number from a uniform distribution between 0 and 3 (inclusive) for the start and end of the episode (same value from episode duration calculation) to account for the midpoint time from subsequent visits.

Generation interval: Difference between the dates of the first positive PCR tests in the index case and in the secondary case each adjusted by adding a random number selected from a uniform distribution between 0 and 3 (inclusive). Restricted to secondary cases who were PCR positive <17 days after the index case. Included mixed subgroup infections as two separate infections.

Results page 12 line 309: The mean duration of shedding was 6.5 days (standard deviation 6.5, range <1-50 days);

Results page 12 lines 311-312: On multivariable analysis, factors associated with longer episode duration were presence of ≥ 2 vs no symptoms, and rRT-PCR Ct value <30 (Table 2, supplementary figure 6).

Supplementary results page 48 lines 918-923: The mean generation interval was 8.4 days (standard deviation 4.0, range 1-16 days) (Supplementary figure 7). On multivariable analysis, factors associated with shorter generation interval were index age group <1, 1-4, 5-12, 13-18 and 19-44 years vs. 45-64 years and contact age group 1-4 years vs 5-12 years (Supplementary table 11). The generation interval was longer for clusters with RSV subgroup B vs. subgroup A.

The change in analysis approach also slightly altered the results of all multivariable models which included duration of shedding as covariates and the model of generation interval. Updated estimates are presented for Table 1, Table 2, Table 3 and Supplementary Table 2.

Could you specify in the methods how you dealt with missed infections (about 10% of samples were missed) and how they were used in calculating the duration of infection/cluster duration/generation interval calculation?

Missed swabs were treated as negative this has been clarified in methods and potential limitations added to discussion.

Methods page 7 line 173: Visits where there was no swab collected were treated as negative.

Discussion page 15 lines 428-429: Missed swabs were treated as negative which could have resulted in underestimation of the duration of infection or HCIR.

Could you provide a bit more information about the statistical methods used? Particularly the Weibull accelerated failure time model; I understand this is an alternative to a proportional hazard model, but unclear why this was chosen. Just a few equations in the SI would help clarify.

*We have updated the supplementary methods as follows page 46 lines 890-899:
We used Weibull accelerated time failure regression to estimate the factors associated with generation interval. Weibull regression implements a fully parametric model (as compared with semi-parametric model implemented by the Cox proportional hazard model).
Advantages of parametric models in survival analysis include: (i) full maximum likelihood can be used to estimate parameters, and (ii) estimated parameters provide clinically meaningful estimates of effect. In Weibull regression the distribution of time to event, T , as a function of single covariate is written as:*

$$\ln(T) = \beta_0 + \beta_1 X + \varepsilon$$

where β_1 is the coefficient for corresponding covariate, ε follows extreme minimum value distribution $G(0, \sigma)$ and σ is the shape parameter.

Also, lines 199-201 are really vague, how were selected predictors varied across models? Also, the forwards backwards selection regression algorithms aren't always the most reliable, could you provide some outputs for which variables were removed at each step?

We have added more detail to the supplement to explain for each model how selected predictors were selected.

Supplementary methods page 46 lines 872-888: Multivariable models were built using forward and backward selection. Age was included in all models a priori. For the model of factors associated with symptoms, age, HIV status and duration of shedding were associated with the outcome on univariate analysis at $p < 0.2$. HIV fell out of the final model because it was confounded by age. For the model of factors associated with episode duration of shedding, minimum Ct value and number of symptoms were associated with the outcome on univariate analysis and retained in the final model. On univariate analysis subgroup untyped was associated with duration of shedding at $p < 0.2$ but this fell out of the final model as it was confounded by cycle threshold value (ie untyped samples had high cycle threshold). For the model of HCIR, number of symptoms and duration of shedding were associated with the outcome on univariate analysis and retained in the final model. Minimum Ct value fell out of the model on multivariable analysis, because of confounding by shedding duration and number of symptoms. Similarly, HIV fell out of the model because of confounding by age. For the model of factors associated with RSV incidence, year and age were associated with the outcome on univariate analysis and retained in the final model. HIV status fell out of the final model because it was confounded by age. For the model of factors associated with generation interval age and subgroup remained in the final model. Underlying illness and duration of shedding were associated on univariate analysis but fell out of the final model as they were confounded by age.

The authors should also test how sensitive these results are to their definition of a new

infection being constrained to >2 weeks from the last day of the previous infection. Given shedding is very long for some individuals, a sensitivity analysis with 1> and >3 weeks would shed light on this.

We have performed the suggested sensitivity analyses to explore the effects of varying the definition of a new infection to >1 and >3 weeks.

Supplementary methods page 46-47 lines 904-906: To explore the effect of our definition of an episode we performed analyses reducing the time between episodes to >1 week and increasing it to >3 weeks.

Supplementary results page 48 lines 935-943: Reducing the interval between episodes to 1 week did not have any effect on the number of episodes or episode duration i.e. number of episodes remained at 400 with duration of 6.7 days as in the original analysis. Increasing the time between episodes to 3 weeks dropped the number of episodes to 393 and increased the mean episode duration to 7.1 days (standard deviation 6.4, range 3-53 days). On this analysis the median generation interval changed marginally to 8.4 days (standard deviation 3.6, range 3-16). Given the small change in episode numbers on these analyses, there was no notable difference in direction and magnitude of factors associated with HCIR, shedding duration or serial interval (data not shown in supplement, results of this analysis, together with analysis code available at https://github.com/crdm-nicd/phirst_rsv).

Figures:

Though Figure 1 is great for getting an overview of the epidemiology, it would be of particular interest to modellers if you could plot histograms of the duration of shedding for significant covariates (i.e. age, and symptoms). Particularly given the large positive skew (e.g. how often are people shedding for longer than 14 days? Seems there are a fair few in Table 2 but these might just be a handful of super-long shedders!)

Thank you for this suggestion. We have added the suggested figures (Supplementary figure 6 a-c) page 89-91.

Figure 1, the y-axis is “cumulative attack rate” no?

Thank you for pointing this out. We have updated the y axis labels.

There also appear to be some people recruited after the peak of incidence, (Figure 1b for example) have the incidence rates been adjusted to account for this?

Yes, rates are estimated accounting for person time in follow up.

Methods page 8 lines 214-215: We defined the incidence of RSV infection or illness as the number of episodes divided by the person time under observation reported per 100 person years.

We have added a discussion of this to limitations page 15 lines 430-432: RSV circulates year-round and we only followed up participants for 10 months of the year, while this period included the months of peak transmission, some cases may have been missed. Additionally, a small number of participants were enrolled after the peak in RSV circulation.

Figure 3a, could you add some grid lines?

Gridlines have been added

Tables in Supp:

Table S1: It's unclear what the difference between the number of rooms for sleeping (1-2[which group is 2 in?], 2-4, >4) and the number of rooms for sleeping (itself) is.

Thanks for picking this up. The correct categories are 1-2, 3-4, >4. This has been corrected.

It's also unclear what the reference is for some of the pK values, (e.g. crowding, children aged >5 years in the household, Female sex etc.) There are a few typos in the captions of this table too.

The table has been updated to include information for the reference groups for all included variable and reviewed and corrected all typos identified.

Table S5: What is A/B? (Perhaps this is in the manuscript, apologies if I missed this.)

The A/B refers to RSV A and RSV B subgroups. Legend has been updated to clarify this.

Table S6 (2020/21 in the caption)? vs Table S7? I'm not quite sure what the difference between these Tables is, could you spell this out a bit more?

The dates have been corrected in the caption of supplementary table 6.

Supplementary table 7 (formerly 6) compares characteristics of participants who were index cases to participants who were never an index case (analysis population all study participants).

Supplementary table 8 (formerly 7) compares characteristics of participants who were index cases to individuals who did have an RSV infection but were not an index case (analysis population all individuals who had an RSV infection episode).

Clarifying language has been added to the table titles.

Supplementary table 7: Factors associated with being an index case of RSV within a household cluster at least once vs not being an index case among 1116 participants in the PHIRST study at a rural and an urban site, South Africa, 2017-2018 (analysis population all study participants)^a

Supplementary table 8: Characteristics associated with being an index case (vs non index case) for 400 infection episodes in household clusters of RSV infection in a rural and an urban site, South Africa, 2017-2018 (analysis population all individuals who had an RSV infection episode)^a

Reviewer #3 (Remarks to the Author):

This is a groundbreaking study. The authors have done a phenomenal job in conducting a prospective household follow-up study of such scale and intensity and report some remarkable findings on the local transmission of RSV in South Africa. Although a study of a similar design was conducted in Kenya over a decade ago, the size and sampling intensity conducted here is truly commendable. The results of the study provide a strong evidence base upon which local policy decisions relating to RSV vaccination can be made.

We thank the reviewer for these positive comments.

Minor points

1. The authors report that the rate of medically-attended infection in infants at 6.1 per 100

persons. Are there similar estimates for older adults over the age of 65? Secondly, because the rates of medically attended infection do not adequately reflect the burden of disease in LMIC settings, did the study collect any objective measures of lower respiratory tract infection in infants and older adults (difficulty in breathing, lower chest wall indrawing, fingertip pulse oximetry or lung function)?

There were no medically attended illness episodes in individuals aged ≥ 65 years, likely because of the low number of included individuals in this age group ($n=45$). This result is presented in supplementary table 3.

We did not collect objective measures of LRTI, this has been added to the limitations section.

Discussion page 14 lines 3829-383: A limitation of our study was that we did not collect data on wheeze or objective measures of lower respiratory tract infection.

2. What are the seasonality patterns of RSV in Mpumalanga Province? Was the follow-up synchronized to match seasonal transmission?

Language describing the timing of the RSV season and the match to the period of follow up has been added to supplementary methods page 42 lines 760-762: To ensure that we captured the start and end of each year's RSV season at each site, twice-weekly follow up was from January through October. The RSV season usually occurs in autumn from February through May and precedes the annual winter influenza season.

3. I was struck by the estimated incidence of illness which was greater in 5-12 years olds (14.9/100pyo) compared to older adults $>65y$ (11.9/100py). Could the authors comment?

In our study we measured mild respiratory illness not requiring medical care. It is not surprising that rates of this type of illness are higher in children aged 5-12 years as this age group has much higher attack rates of infection (48.7 vs 17.8 episodes per hundred person years), likely because of higher contact patterns and possibly lower levels of immunity compared to adults who have had many more previous infections. As expected symptomatic fraction was higher in the elderly (67%) vs 5-12 years (37%). The difference in symptomatic illness rates between these two groups therefore seems to be driven by the high infection attack rates in young individuals.

4. At least one participant was reported to have shed virus for 53 days. Shedding duration was reportedly associated with age, symptoms and viral load. Could this also potentially be explained by HIV status

On multivariable analysis, PLWH did not have longer shedding compared to HIV-uninfected individuals (Table 2). We have added a sentence to the limitations sections to indicate due to limited power we cannot exclude associations with HIV.

Discussion page 13 lines 363-366: An important limitation of this analysis was the relatively small numbers of PLWH included in the study which may have limited power to detect associations. We were also not able to compare the characteristics of PLWH well-controlled on treatment to those poorly controlled.

REVIEWERS' COMMENTS

Reviewer #1 (Remarks to the Author):

I have reviewed the revised manuscript and the reaction of the authors to the comments raised. All my comments have been adequately addressed and the manuscript has been improved substantially. I have no further comments.

Reviewer #2 (Remarks to the Author):

The authors of "Incidence and transmission of respiratory syncytial virus in urban and rural communities in South Africa, 2017-2018: results of the PHIRST cohort study" have responded to all my queries comprehensively and made the appropriate changes throughout the manuscript and Supplementary Information. I. have no further comments.